# Differences in the Effects of Work Environment on Health Problems and Satisfaction of Working Condition by Gender: The 6th Korean Working Conditions Survey

**DOI:** 10.3390/ijerph20196824

**Published:** 2023-09-25

**Authors:** Chae Hyeseon, Park Sooin, Kim Insoo, Ko Myungsun

**Affiliations:** Rural Development Administration, Jeonju-si 54875, Republic of Korea; hyeseon@korea.kr (C.H.); sooinpark@korea.kr (P.S.); ergonomist@korea.kr (K.I.)

**Keywords:** work-related, working condition, farmworker, agriculture, musculoskeletal, KWCS

## Abstract

This study investigated gender differences in exposure to hazardous factors, health problems, and work environment satisfaction and identified the effects of such differences on farmworkers in Korea. Through the 6th Korean Working Conditions Survey (KWCS) conducted by the Occupational Safety and Health Research Institute (OSHRI), the raw data on 2347 farmworkers were analyzed to present descriptive statistics for demographic characteristics, exposure to hazardous factors, health problems, and work environment satisfaction. We compared genders using chi-squared tests and investigated the effects of gender-specific working conditions on work-related health problems and work environment satisfaction using multivariate logistic analysis. The results were presented as an odds ratio (OR) with 95% confidence interval. Job position predicted work-related health problems in male farmworkers and work environment satisfaction in female farmworkers. Furthermore, female farmworkers perceived themselves to have more health problems than male farmworkers. Nevertheless, female farmworkers received less health and safety information and had lower work environment satisfaction compared to male farmworkers. The findings may improve the occupational welfare of farmworkers through continuous enhancement of the agricultural labor environment.

## 1. Introduction

The agricultural sector in Korea confronts a multitude of obstacles, including a higher incidence of accidents compared to other industries, a progressively aging and predominantly female workforce, and the pressing demand to secure a labor force that enables sustainable agricultural management. The injury rate in the agricultural sector stands at 1.4 times higher than the average injury rate for all industries [1], with agricultural workers exposed to an elevated level of hazards and susceptible to a greater prevalence of work-related musculoskeletal pain compared to the general workforce [2]. For instance, occupational disorders among farmworkers are musculoskeletal disorders, skin diseases, pesticide poisoning, infectious diseases, respiratory diseases, and diseases incurred by exposure to noise and hot and cold temperatures [3]. Moreover, in 2020, the proportion of adults aged 65 and older among the farming population was roughly three times higher than that in the overall elderly population, and it is steadily rising.

As of 2021, the agriculture, forestry, and fisheries sector accounts for a meager 0.2% of all industries and 0.3% of the total workforce [4]. Nevertheless, there are approximately 1.7 million agricultural businesses, which comprise agricultural business proprietors and agricultural corporations that have registered agricultural management information [5]. Most farmers are self-employed, and about 67% of agricultural corporations are small businesses with less than five full-time workers [6], rendering agriculture an industry prone to high rates of safety incidents with a weak safety and health management system. Similar to this finding, the Industrial Accidents Report [1] shows that industrial accidents occur more frequently in small to medium-sized workplaces with fewer full-time workers.

However, the proportion of farm owners has increased in recent years due to a rise in the number of new farmers, including young adults, female farmers, and those transitioning from other industries [5]. The number of agricultural corporations has also increased, owing to agricultural business benefits such as corporate tax reduction. Furthermore, the foreign work permit system was introduced to ensure sustainable agricultural management and address the labor shortage issue and is being utilized actively in the agricultural sector. Nonetheless, foreign seasonal workers are subject to poor working conditions, including various hazardous tasks, calling for improved working conditions and labor welfare.

In 2020, female farm owners accounted for approximately 20% of all farm owners [7], and this percentage is rapidly increasing along with the growing influx of female farm owners from other industries who transitioned to agriculture [5]. As the roles and importance of female farmers in the agricultural and rural communities continue to grow, such as in the handling of agricultural machinery and active involvement in farming activities, policies and programs aimed at reducing labor and improving health and safety have been implemented. Examples of such initiatives are programs by the Ministry of Agriculture, Food and Rural Affairs (MAFRA), which provide farm helpers with education to improve farming conditions and assistance for female-friendly, convenient farming equipment. In 2022, a pilot project for special health examination for female farmers was introduced to prevent agricultural work-related diseases and promote health and well-being. Additionally, the Act on Safety Insurance for Farmers and Fishers and Prevention of Work Accidents has introduced new regulations pertaining to the implementation of programs for the prevention of safety accidents in agricultural and fishery industries.

To advance various research and development endeavors and strengthen policymaking, it is important to examine the gender differences in farming work environments and identify their associated factors. Men and women have differences in physical characteristics, and their work characteristics and conditions may also differ, resulting in different trends in the incidence, type, and cause of work-related diseases and injuries. The physical work capacity of women is about 70% of men’s, and there are also differences in their physiology, such as muscular strength and oxygen consumption [8]. Female workers report more health problems related to fatigue, repetitive strain injury, infectious diseases, musculoskeletal disorders, and mental distress than male workers [9,10]. In the agricultural sector, women experience more musculoskeletal pain or disorders than men [2,11], and given that musculoskeletal disorders have a slow onset over prolonged periods of strain, it is difficult to prove their relation to work conditions. There are also gender differences in agricultural work-related injuries, where female farmers experience more injuries due to “falls” and “excessive force/movement”, while male farmers report more injuries due to “agricultural machinery”, “collision with objects”, “entanglement/pinching”, and “animal injuries” [11]. Kim (2010) [12] also reported that female farmers experience more work-related musculoskeletal disorders than male farmers. The Rural Development Administration [11] reports that male farmers are more exposed to noise, exhaust gas, and handling of heavy objects, while female farmers experience more musculoskeletal symptoms and visual impairment than male farmers. Findings and reports confirm gender differences in the occurrence of work-related accidents and exposure to hazardous factors in agriculture.

However, there is a lack of research on gender differences in work environment satisfaction in the agricultural sector, which hinders the understanding of the gender differences in the effects of agricultural work conditions on health problems and work environment satisfaction.

This study investigates the gender-specific exposure to hazardous factors, health problems, and work environment satisfaction among farmworkers and identifies the gender-specific predictors of work-related health problems and work environment satisfaction. Using data on farmworkers in the 6th Korean Working Conditions Survey (KWCS) conducted by the Occupational Safety and Health Research Institute (OSHRI), we ultimately aspire to present foundational data to enhance agricultural working conditions and facilitate relevant research.

## 2. Materials and Methods

### 2.1. Data Source and Participants

The raw data from the 2020 (6th) KWCS conducted by the OSHRI were used. The KWCS is conducted every three years on 50,000 economically active Korean residents aged 15 years and older. Economically active individuals are defined as “individuals who have worked at least one hour for income in the past week”, which includes “self-employed individuals without any employees”, “self-employed individuals and business owners with employees”, “wage workers”, “unpaid family workers”, and “workers on temporary leave”. The survey contains about 130 items pertaining to worker characteristics (e.g., job position, exposure to hazardous factors, labor intensity, job stress, emotional labor, violence and discrimination, and work hours.) The participants were sampled via stratified sampling in consideration of region, size of region, and type of housing to establish a sample representative of the farmworker population in Korea.

From 50,538 participants of the 6th KWCS, we analyzed data on 2347 participants classified as farmworkers. In accordance with the KSCO or Korean Standard Classification of Occupations (7th version, 3 July 2017), farmworkers were defined as individuals engaged in crop cultivation (90.7%), horticulture and landscaping (4.0%), and animal husbandry and livestock-related work (5.3%).

### 2.2. Definition and Measurement of Variables

Variables unrelated to agricultural field were excluded from the items on the 6th KWCS. We selected the following variables appropriate for investigating agricultural working conditions. Gender and age were used as demographic characteristics, while job position, duration of farming career, exposure to hazardous factors, work-related health problems, and work environment satisfaction were selected to analyze gender-specific working conditions.

#### 2.2.1. Exposure to Hazardous Factors

Questions Q28 and Q29 were used to survey exposure to hazardous factors among farmworkers. For Q28, data on exposure to “vibration”, “noise”, “high temperature”, “low temperature”, “inhalation of smoke/fume/power/dust”, and “handling of chemical products/substances” were used. For Q29, data on exposure to “postures that cause fatigue or pain (excluding postures of continuous standing or sitting)”, “pulling/pushing/moving heavy objects”, “continuous standing”, “sitting posture”, and “repetitive hand/arm movements” were used. The responses “all throughout work hours”, “during most of work hours”, “during three-quarters of the work hours”, “during half of the work hours”, and “during one-quarter of work hours” were defined as “exposure to the corresponding hazardous factors”. Meanwhile, the responses “almost no exposure at all” and “absolutely no exposure” were defined as “no exposure to the corresponding hazardous factors”.

#### 2.2.2. Perceived Health Problems and Their Relevance to Work

Q70 of the 6th KWCS asks about participants’ perceived health problems in the past year (or after commencement of work if the participant has not yet worked for a full year). Participants’ perceived “low back pain”, “upper extremity muscle pain (shoulder/neck/arm/elbow/hand)”, “lower extremity muscle pain (hip/legs/knees/feet)”, “headache/eye strain”, and “anxiety” were coded as 1 for the response “no” or 2 for “yes”. Q71-1 asks about their perceptions about whether these health problems are work-related, and the responses were also coded as 1 for “no” or 2 for “yes”.

#### 2.2.3. Availability of Health-and-Safety-Related Information

Q30 of the 6th KWCS asks about whether participants receive “information about factors that threaten health and safety regarding their work”. “Factors that threaten health and safety” refers to various work-related hazards that may potentially cause mental and physical harm. The responses “I am given adequate information” and “I am given adequate information in general” were coded as “given information”. The responses “I am not really given much information” and “I am not given any information at all” were coded as “not given information”.

#### 2.2.4. Work Environment Satisfaction

Q77 of the 6th KWCS, which asks about “overall satisfaction with the work environment”, was used to investigate work environment satisfaction. The response options in the KWCS are “very satisfied”, “satisfied”, “dissatisfied”, “very dissatisfied”, but we changed the coding to “satisfied” and “dissatisfied”. The responses “very satisfied” and “satisfied” were defined as “satisfied”, while “dissatisfied” and “very dissatisfied” were defined as “dissatisfied”.

### 2.3. Statistical Analysis

Statistical analyses were performed using the SPSS ver. 18.0 software. Demographic characteristics, exposure to hazardous factors, health problems, and work environment satisfaction were analyzed using descriptive statistics, while gender-specific differences were analyzed through chi-squared tests. Furthermore, the effects of working conditions on work-related health problems and work environment satisfaction in male and female farmworkers were analyzed using multivariate logistic regression and presented as an odds ratio (OR) and 95% confidence interval (CI)

## 3. Results

### 3.1. Participants’ Demographic Characteristics

Table 1 shows the demographic characteristics of 2347 farmworkers by gender (62.8% male, 37.2% female). Comparisons between the two groups were made using *t*-test. Among men, 44.2% were under 60 years old, and 35.0% were age 65 or older. Among women, 40.8% were under 60 years old, and 59.2% were age 65 or older. There were no significant differences in the age groups between male and female farmworkers. As for job position, there were more “self-employed farmers without employees” among men (73.9%) than women (47.5%), and there were more “self-employed farmers/business owners with employees” among men (21.7%) than women (10.8%). On the other hand, there were more “wage earners (employees)” among women (3.8%) than men (2.9%), and particularly, there was a markedly higher percentage of “unpaid family workers” among women (38.0%) than men (1.5%). Regarding length of career, there were statistically significant gender differences in the percentages of workers with “less than 10 years of farming career” (male: 15.6% vs. female: 15.4%), “between 10–29 years of farming career” (male: 31.8% vs. female: 24.5%), and “≥30 years of farming career” (male: 52.6% vs. female: 60.1%).

### 3.2. Exposure to Hazardous Factors among Farmworkers by Gender

Table 2 shows the gender differences in exposure to hazardous factors among farmworkers. The percentages of male and female farmworkers exposed to “low temperature (indoor/outdoor)” were 30.1% and 29.7%, respectively, and those exposed to “high temperature causing sweating even when not working” were 40.8% and 44.0%, respectively. Further, the percentages of male and female workers exposed to “severe noise that requires a loud voice when talking to others” were 11.1% and 10.6%, respectively. These gender gaps were not statistically significant.

On the other hand, there was a significantly higher percentage of men exposed to “vibration generated by hand tools/machinery” (32.1%) than women (17.5%).

There were no significant differences for “handling/contact with chemical products/substances” between men (11.0%) and women (9.2%). However, a significantly higher percentage of men (20.5%) were exposed to “inhalation of smoke/fume/powder/dust” than women (15.0%).

Regarding ergonomic risks, there were no significant differences in exposure to “repetitive hand/arm movement” between men (80.5%) and women (81.6%), but exposure to other ergonomic risk factors significantly differed between genders. Women (81.0%) were more exposed to “sitting posture” than men (69.0%), while men (89.2%) were more exposed to “continuous standing posture” than women (84.3%). “Pulling/pushing/moving heavy objects” was more common among men (69.4%) than women (22.0%), while “posture that causes fatigue or pain” was more common among women (70.5%) than men (64.2%).

### 3.3. Perceived Health Problems among Farmworkers by Gender

Table 3 shows the gender gaps in perceived health problems in the past year among farmworkers. The percentages of farmworkers experiencing low back pain, upper extremity pain, lower extremity pain, headache/eye strain, and anxiety were significantly higher for female workers than male workers. Low back pain was present in 56.1% of men and 74.4% of women, and upper extremity pain was present in 56.4% of men and 68.6% of women. Lower extremity pain was presented in 43.9% of men and 61.4% of women, and headache/eye strain was present in 15.5% of men and 20.4% of women. Anxiety was present in 3.3% of men and 6.6% of women.

### 3.4. Availability of Health-and-Safety-Related Information among Farmworkers by Gender

Regarding the availability of health-and-safety-related information for farmworkers by gender (Table 4), 67.9% of men and 75.8% of women stated that they were not given information about health and safety, and the percentages significantly differed between the two groups.

### 3.5. Work Environment Satisfaction among Farmworkers by Gender

Gender differences in work environment satisfaction among farmworkers were analyzed (Table 5). The percentage of farmworkers who are satisfied with their work environment was significantly higher in men (77.9%) than women (72.0%).

### 3.6. Predictors of Work-Related Health Problems and Work Environment Satisfaction among Farmworkers by Gender

To analyze the effects of work conditions on work-related health problems among farmworkers by gender, we performed a multivariate logistic regression with presence of work-related health problems as the dependent variable. Job position, length of career, exposure to hazardous factors, and availability of health and safety information were assigned as the independent variables (Table 6).

Among male farmworkers, self-employed farmers/business owners with employees had 3.298 times (95% CI 1.479∼7.353) higher risk of low back pain, 4.062 times (95% CI 1.828∼9.028) higher risk of upper extremity pain, 17.366 times (95% CI 4.703∼64.121) higher risk of lower extremity pain, and 5.153 times (95% CI 1.700∼15.620) higher risk of headache/eye strain compared to farmers/business owners without employees. Among female farmworkers, wage earners (employees) had 0.268% lower odds (95% CI 0.087∼0.827) of acquiring low back pain than those who were self-employed without employees.

The length of career was associated with the risk of developing work-related health problems in both male and female. Those with a career of ≥30 years were 3~6 times more likely to develop musculoskeletal symptoms compared to those with <10 years of career.

Regarding exposure to hazardous factors in male farmworkers, the risk factors for low back pain were: high temperature (OR = 1.907 [95% CI 1.052∼3.457]), contact with/handling of chemical substances (OR = 3.086 [95% CI 1.061∼8.974]), pulling/pushing/moving heavy objects (OR = 1.961 [95% CI 1.132∼3.396], and repetitive hand/arm movement (OR = 3.278 [95% CI 1.887∼5.693]). The risk of upper extremity pain was 3.322 times higher (95% CI 1.876∼5.885) for those who handle heavy objects, 4.254 times (95% CI 2.354∼7.696) for those with exposure to repetitive hand/arm movement, and 2.799 times (95% CI 1.642∼4.770) for those not given health-and-safety-related information. The OR for lower extremity muscle pain was 2.721 times (95% CI 1.460∼5.070) for those who handle heavy objects, 0.279 times (95% CI 0.124∼0.625) for those who work in a sitting posture, 4.242 times (95% CI 2.183∼8.242) for those who engage in repetitive hand/arm movement, and 2.036 times (95% CI 1.097∼3.782) for those who are not given health-and-safety-related information. The risk for work-related headache/eye strain was 2.774 times higher (95% CI 1.102∼6.985) for those who engage in repetitive hand/arm movement. The risk for anxiety was 5.071 times higher (95% CI 0.945∼27.223) for those working in a posture that causes fatigue/pain and 11.371 (95% CI 2.224∼58.138) for those who handle heavy objects.

In female farmworkers, the risk of low back pain was 2.105 times higher (95% CI 1.090∼4.066) for those who work in a continuous standing posture, 2.289 times (95% CI 1.197∼4.378) for those who engage in repetitive hand/arm movement, and 1.892 times (95% CI 1.029∼3.479) for those who were not given health-and-safety-related information. The risk of upper extremity muscle pain was 2.874 times higher (95% CI 1.499∼5.508) for those who handle heavy objects, and 2.018 times (95% CI 0.989∼4.118) for those who were not given health-and-safety-related information. The risk of lower extremity muscle pain was 2.739 times higher (95% CI 1.292∼5.806) for those who engage in repetitive hand/arm movement. The risk of headache/eye strain was 2.195 times higher (95% CI 1.091∼4.415 for those who handle heavy objects and 2.260 times (95% CI 0.973∼5.250) for those not given health-and-safety-related information. The risk for anxiety was 4.358 times higher (95% CI 1.189∼15.975) for those who handle heavy objects.

In terms of work environment satisfaction among male farmworkers, the OR was 0.487 (95% CI 0.322∼0.736) for those with ≥30 years of career compared to those with <10 years of career, 0.748 (95% CI 0.571∼0.981) for those exposed to high temperatures, 0.739 (95% CI 0.551∼0.990) for those working in a sitting posture, and 0.312 (95% CI 0.229∼0.450) for those not given health-and-safety-related information. Among female farmworkers, the OR for work environment satisfaction was 3.090 (95% CI 1.481∼6.449) for self-employed farmers/business owners with employees compared to self-employed farmers without employees, 0.354 (95% CI 0.198∼0.631) for those with ≥30 years of career compared to those with <10 years of career, and 0.554 (95% CI 0.368∼0.834) for those not given health-and-safety-related information.

## 4. Discussion

We investigated the gender differences in exposure to hazardous factors, health problems, and work environment satisfaction. Also, we identified the effects of gender-specific working conditions on work-related health problems and work environment satisfaction among farmworkers using data from the 6th KWCS conducted by the OSHRI.

In our study, job position affected work-related health problems in men and work environment satisfaction in women. Among male farmworkers, the odds for work-related low back pain (3.298 times), upper extremity pain (4.062 times), lower extremity pain (17.366 times), and headache/eye strain (5.153 times) were higher among the self-employed/business owners with employees, compared to the self-employed without employees. Moreover, Lee (2017) [13] reported that job position is an occupational characteristic that predicts self-rated health status among workers, which is consistent with our results for male farmworkers; however, job position did not affect work-related health problems in female farmworkers in our study. Among female farmworkers, the odds for being satisfied with the work environment increased by 3.090 times among the self-employed/business owners with employees, compared to the self-employed without employees. This is because farms with employees, including short-term employees, only account for 37% of all farms [7]; and even in farms with employees, male farm owners experience an increased burden of farm management and labor due to the farm labor structure; where timely response to labor demands is hindered by labor demand being concentrated in the busy season.

Furthermore, previous studies have reported that job position and years of working are occupational characteristics that predict self-rated health status [13] and that work injuries increase along with the length of farming career [14]. In our study, the odds for musculoskeletal pain and headache/eye strain increased along with the length of farming career regardless of gender, and work environment satisfaction decreased with the increasing length of career. While not many studies have examined work-related headache/eye strain among farmers, some studies have shown that handling of pesticides and farming machinery serve as triggers of headache. Shala Chetty-Mhlanga et al. (2021) [15] reported that farming activities involving pesticides are associated with the onset of headache, and S. Scutter (1997) [16] showed that more than 70% of farmers in Australia experience neck pain and headache caused by vibration sustained by the whole body and neck rotation from driving a tractor. D. Villarejo [17] reported that some pesticide-exposure-related health problems among farmworkers include blurred vision and headache.

In our study, we observed that there are gender differences in the exposure to hazardous factors. Men were more likely to be exposed to vibration, inhaled smoke/fumes/powder/dust, continuous standing posture, and pulling/pushing/moving heavy objects than women. On the other hand, women were more likely to be exposed to sitting posture and postures that cause fatigue or pain. A farmer survey conducted by the Rural Development Administration [11] also reported gender differences in exposure to hazardous factors, where male farmers are more frequently exposed to noise, exhaust gas, and handling of heavy objects, while female farmers are more frequently exposed to knee/low back musculoskeletal risk factors.

In our study, male and female farmworkers also significantly differed in their perceived health problems. Women perceived more musculoskeletal pain (low back pain, upper extremity muscle pain, lower extremity muscle pain), and headache/eye strain than men. Our results for musculoskeletal pain were consistent with previous reports that the prevalence of musculoskeletal disorders is higher among female farmers than male farmers [2,11,12]. However, these results were contradictory to a previous report on workers with approved work compensation [18], where musculoskeletal disorder was more prevalent among men than women, and to a study by Jang & Lee (2021) [19], where depression and anxiety were more prevalent among men than women. These results are attributed to the differences in work environments between general industries and agriculture and the consequent differences in the trends of common health problems per industry.

Female farmworkers were also found to be less satisfied with their work environment than male farmworkers, which is inconsistent with the findings of Kim et al. (2010) [12], where there was no gender gap in job satisfaction among farmers. In the present study, exposure to high temperatures during work contributed to lower work environment satisfaction among male farmworkers despite the lack of significant differences in exposure to high temperature between male and female farmworkers. This may be attributable to gender discrepancies in sweat loss and body temperature in response to hot environments [20] and the possibility that men are more vulnerable in terms of their physical adaptation and burden in response to thermal stress [21]. Therefore, to improve work environment satisfaction, a safety and health management system must be established to manage diverse hazardous factors, ensure adequate rest, and provide sufficient information about safety and health.

We further observed significant gender differences in exposure to hazardous factors that influence work-related health problems among farmworkers. In men, high temperature, contact with/handling of chemical substances, pulling/pushing/moving heavy objects, and repetitive hand/arm movement affected low back pain, but, in women, continuous standing posture and repetitive hand/arm movement affected low back pain. Meanwhile, upper extremity pain was affected by pulling/pushing/moving heavy objects and repetitive hand/arm movement, with no significant differences between the gender groups. Lower extremity pain was affected by pulling/pushing/moving heavy objects and repetitive hand/arm movement in men and repetitive hand/arm movement in women. The predictors of headache/eye strain also significantly differed between genders, which were repetitive hand/arm movement for men and pulling/pushing/moving heavy objects for women. Although there is a dearth of studies that can be compared with our findings on the predictors of work-related health problems among farmworkers, Kim et al. (2010) [12] reported that major types of work activities differ between genders, but work intensity does not differ between genders among farmers. Hence, it is necessary to examine the level of risk exposure in the workplace, define appropriate exposure limits, and develop safety procedures [22]. Additionally, it is important to implement measures to improve the working environment, such as developing standard operating procedures that consider gender-specific physical characteristics and work activities.

Additionally, although female agricultural workers in this study were found to experience more health problems than their male counterparts, they received less health-and-safety-related information and exhibited lower work environment satisfaction. Furthermore, both male and female farmworkers who did not receive health and safety information were less satisfied with their work environment. Between the two groups, the odds for work-related musculoskeletal pain were about twofold higher among those who did not receive health-and-safety-related information in men, while the odds for work-related headache and eye strain were about twofold higher among those without safety-and-health-related information in women. Despite being exposed to more hazardous factors than general workers, farmworkers receive less information about health and safety [23]. Businesses with fewer than five full-time employees are exempt from industrial safety and health management system regulations, safety and health management regulations, and regulations for additional education [24]. Moreover, the mandatory appointment of safety and health management personnel in small and medium-sized workplaces does not apply to the agricultural field [24]. Therefore, to enhance the working environment and prevent industrial accidents in agriculture, it is essential to strengthen the safety and health management systems of small-sized farms, which currently face significant gaps in terms of safety management. Regarding the health of female farmers, there is a pressing need to assess health risk factors and measures to manage them, provide farming assistance during agricultural accidents, and expand the availability of convenient farming equipment and personal protective equipment [25]. Moreover, a practical safety and health education system tailored to agricultural workplaces also needs to be implemented.

Meanwhile, the only mental health issue that can be identified through KWCS data is anxiety, and, in this study, the prevalence of work-related anxiety was higher among female farmworkers than male farmworkers (contradictory to some previous findings [25,26,27,28].) B. Sanne et al. (2004) [27] found that farmers face a higher incidence and severity of worrying and depression than non-farmers, with male farmers experiencing higher levels of worrying and depression than female farmers, presumably due to long working hours, physically demanding tasks, and low income. Likewise, Andria Jones-Bitton (2020) [28] reported that farmers display a high level of stress, anxiety, and depression with low resilience, and that these factors impair mental health more in men than women. Based on our study, the predictors of work-related anxiety in male farmworkers are postures that cause fatigue or pain (5.071 times higher risk) and pulling/pushing/moving heavy objects (11.371 times), while, for female farmworkers, it is pulling/pushing/moving heavy objects (4.358 times higher). In other words, handling heavy objects was a common risk factor for work-related anxiety in both gender groups. An interesting point is that handling heavy objects was a more potent predictor of work-related anxiety in men (about 11 times higher) than women (about 4 times) despite the significantly higher prevalence of anxiety in women than men.

Gong (2007) [29] also suggested that various work conditions, such as job characteristics and urgent responses to accidental situations, are predictors of work-related anxiety, while Jang & Lee (2021) [19] identified the risk for musculoskeletal disorders as a predictor of work-related anxiety. On the contrary, Iveth Cuellar Celallos (2022) [30] claimed that high-intensity physical pain causes depression and anxiety in farmworkers. In the present study, we observed that handling heavy objects and postures that cause fatigue/pain were strong predictors of work-related anxiety, which is consistent with previous research findings.

Previous research has also discovered the following on farmers’ mental health: older women, older adults without a spouse, and economically vulnerable older adults display a higher level of depression and anxiety [31,32], and middle-aged and older workers show increased severity of depression with increasing perceived physically demanding job demands [33]. Further, depression and anxiety are strongly correlated [19]. Exposure to pesticides is also correlated with mental health, and the correlation between pesticide exposure and depressive mood is more evident in women than men [34]. Health problems in workers contribute to reduced work productivity [35], and job stress also influences workers’ work capacity [36], calling for mental health management in workplaces.

Overall, there is inadequate research and discussion on anxiety and its underlying causes among farmworkers. A comprehensive understanding of the level of anxiety and the multifaceted personal, social, and occupational factors contributing to anxiety among farmworkers is crucial to explore potential solutions for improving their mental health.

The use of the KWCS data for the present study brings about some limitations. First, we were unable to precisely identify ergonomic risk factors intrinsic to farming activities, such as crouching, arm-raising, and bending over. Second, work-related health problems were limited to musculoskeletal pain in the low back and the upper and lower extremities as well as eye strain and headaches, and we used anxiety only as an indicator of mental health. Third, there is a lack of survey data that shed light on the labor environment for foreign seasonal workers, a population with increasing presence in the agricultural sector. Lastly, the KWCS utilized in this study is intended to investigate the standardized working conditions in various industries. Since the health problems that could be relevant for the entire working population were selected, we had limitations in identifying health problems that are specific to the agricultural field.

Nevertheless, a key strength of this study is that the findings can be generalized to the entire farmworker population in Korea, as we analyzed nationally representative sample data for the Korean working population. Another strength is that we confirmed that there are gender differences in working conditions, health problems, and work environment satisfaction among farmworkers and identified the specific working conditions that predict work-related health problems and work environment satisfaction.

In the future, standardized research and surveys are needed to examine hazardous factors specific to different types of farming jobs and identify predictors of work-related health problems and occupational welfare. Such findings will be useful for enhancing occupational welfare for farmworkers through continuous improvement of agricultural work environments.

## 5. Conclusions

Regardless of gender, work-related musculoskeletal pain increased while work environment satisfaction decreased with increasing length of career among farmworkers. Moreover, there were gender differences in exposure to hazardous factors, perceived health problems, and work environment satisfaction among farmworkers. Female farmworkers were found to be given less health-and-safety-related information, displayed lower work environment satisfaction, and experienced more health problems compared to their male counterparts.

## Figures and Tables

**Table 1 ijerph-20-06824-t001:** Demographic characteristics of farmworkers by gender.

Variables	Male		Female		x^2^	*p*-Value
	*n*	%	*n*	%		
**Age** (Total)	1473	100.0	874	100.0	2.511	0.113
<60 Years	651	44.2	357	40.8		
≥65 Years	558	35.0	517	59.2		
**Job position** (Total)	1473	100.0	874	100.0	583.069	0.000 ***
Self-employed without employees	1089	73.9	415	47.5		
Self-employed/business owners with employees	320	21.7	94	10.8		
Wage earners (employees)	42	2.9	33	3.8		
Unpaid family workers	22	1.5	332	38.0		
**Length of career** (Total)	1473	100.0	874	100.0	15.435	0.000 ***
<10 Years	230	15.6	135	15.4		
10–29 years	468	31.8	214	24.5		
≥30 years	776	52.6	525	60.1		

*** *p* < 0.001.

**Table 2 ijerph-20-06824-t002:** Exposure to hazardous factors among farmworkers by gender.

Variables	Male		Female		x^2^	*p*-Value
	*n*	%	*n*	%		
**Low temperature (indoor/outdoor)** **(** **Total** **)**	1472	100.0	872	100.0	0.040	0.841
Almost none or none at all	1029	69.9	613	70.3		
More than ¼ of work hours	443	30.1	259	29.7		
**High temperature causing sweating even when not working** **(** **Total** **)**	1472	100.0	872	100.0	2.413	0.120
Almost none or none at all	872	59.2	488	56.0		
More than ¼ of work hours	600	40.8	384	44.0		
**Severe noise that requires a loud voice when talking to others** **(** **Total** **)**	1472	100.0	872	100.0	0.196	0.658
Almost none or none at all	1308	88.9	780	89.4		
More than ¼ of work hours	164	11.1	92	10.6		
**Vibration generated by hand tools and machinery** **(** **Total** **)**	1472	100.0	872	100.0	59.530	0.000 ***
Almost none or none at all	999	67.9	719	82.5		
More than ¼ of work hours	473	32.1	153	17.5		
**Handling/contact with chemical products/substances** **(** **Total** **)**	1469	100.0	872	100.0	1.889	0.169
Almost none or none at all	1308	89.0	792	90.8		
More than ¼ of work hours	161	11.0	80	9.2		
**Inhalation of smoke, fume, powder, or dust** **(** **Total** **)**	1471	100.0	872	100.0	10.771	0.001 ***
Almost none or none at all	1170	79.5	741	85.0		
More than ¼ of work hours	301	20.5	131	15.0		
**Repetitive hand/arm movement (Total)**	1474	100.0	874	100.0	0.392	0.531
Almost none or none at all	287	19.5	161	18.4		
More than ¼ of work hours	1187	80.5	713	81.6		
**Sitting posture (Total)**	1473	100.0	874	100.0	40.726	0.000 ***
Almost none or none at all	457	31.0	166	19.0		
More than ¼ of work hours	1016	69.0	708	81.0		
**Continuous standing posture (Total)**	1473	100.0	874	100.0	11.856	0.001 ***
Almost none or none at all	159	10.8	137	15.7		
More than ¼ of work hours	1314	89.2	737	84.3		
**Pulling/pushing/moving heavy objects (Total)**	1474	100.0	873	100.0	25.194	0.000 ***
Almost none or none at all	451	30.6	356	40.8		
More than ¼ of work hours	1023	69.4	517	22.0		
**Posture that causes fatigue or pain (Total)**	1473	100.0	874	100.0	9.650	0.002 **
Almost none or none at all	527	35.8	258	29.5		
More than ¼ of work hours	946	64.2	616	70.5		

** *p* < 0.01, *** *p* < 0.001.

**Table 3 ijerph-20-06824-t003:** Perceived health problems in farmworkers by gender.

Variables	Male		Female		x^2^	*p*-Value
	*n*	%	*n*	%		
**Low back pain** **(** **Total** **)**	1473	100.0	874	100.0	78.663	0.000 ***
No	647	43.9	224	25.6		
Yes	826	56.1	650	74.4		
**Upper extremity Muscle pain (Total** **)**	1472	100.0	873	100.0	33.325	0.000 ***
No	639	43.4	274	31.4		
Yes	833	56.4	599	68.6		
**Lower extremity muscle pain** **(** **Total** **)**	1473	100.0	873	100.0	67.469	0.000 ***
No	827	56.1	337	38.6	15.435	0.000 ***
Yes	646	43.9	536	61.4		
**Headache** **,** **Eye strain** **(** **Total** **)**	1473	100.0	874	100.0	9.159	0.002 **
No	1245	84.5	696	79.6		
Yes	228	15.5	178	20.4		
**Anxiety** **(** **Total** **)**	1473	100.0	873	100.0	13.857	0.000 ***
No	1424	96.7	815	934		
Yes	49	3.3	58	6.6		

** *p* < 0.01, *** *p* < 0.001.

**Table 4 ijerph-20-06824-t004:** Availability of health-and-safety-related information for farmworkers by gender.

Variables	Male		Female		x^2^	*p*-Value
	*n*	%	*n*	%		
**Availability of health & safety information (** **Total)**	1449	100.0	856	100.0	16.296	0.000 ***
Given information	465	32.1	207	24.2		
Not given information	984	67.9	649	75.8		

*** *p* < 0.001.

**Table 5 ijerph-20-06824-t005:** Work environment satisfaction among farmworkers by gender.

Variables	Male		Female		x^2^	*p*-Value
	*n*	%	*n*	%		
**Work environment satisfaction (Total)**	1473	100.0	874	100.0	10.373	0.000 ***
Satisfied	1147	77.9	629	72.0		
Not satisfied	326	22.1	245	28.0		

*** *p* < 0.001.

**Table 6 ijerph-20-06824-t006:** Predictors of work-related health problems and work environment satisfaction among farmworkers.

Variables	Low Back Pain		Upper Extremity Pain		Lower Extremity Pain		Headache/Eye Strain		Anxiety		Work Environment Satisfaction	
	Male	Female	Male	Female	Male	Female	Male	Female	Male	Female	Male	Female
	OR (95% CI)	OR (95% CI)	OR (95% CI)	OR (95% CI)	OR (95% CI)	OR (95% CI)	OR (95% CI)	OR (95% CI)	OR (95% CI)	OR (95% CI)	OR (95% CI)	OR (95% CI)
**Job position**												
Self-employed without employees	1	1	1	1	1	1	1	1				1
Self-employed/business owners with employees	3.298 (1.479~7.353) **	3.272 (0.934~11.463)	4.062 (1.828~9.028) ***	2.176 × 10^8^ (0.000)	17.366 (4.703~64.121) ***	2.221×10^8^ (0.000)	5.153 (1.700~15.620) **	7.706×10^8^ (0.000)				3.090 (1.481~6.449) **
Wage earners (employees)	1.785 (0.270~11.809)	0.268 (0.087~0.827) *	1.717 (0.265~11.114)	0.350 (0.084~1.451)	0.549 (0.072~4.153)	0.476 (0.095~2.392)	6.481×10^8^ (0.000)	0.293 (0.015~5.913)				0.664 (0.240~1.841)
Unpaid family workers	0.212 (0.049~0.916) *	1.508 (0.810~2.806)	0.513 (0.097~2.714)	1.592 (0.818~3.101)	0.246 (0.054~1.117)	1.438 (0.719~2.876)	0.206 (0.030~1.422)	1.973 (0.938~4.152)				1.003 (0.722~1.393)
**Career duration**												
<10 years	1		1	1	1	1	1	1			1	1
10~29 years	2.641 (1.153~6.049) *		1.461 (0.617~3.461)	2.005 (0.799~5.030)	1.523 (0.546~4.249)	2.635 (0.909~7.643)	1.249 (0.380~4.104)	5.166 (1.176~22.684) *			0.664 (0.424~1.041)	0.737 (0.386~1.408)
≥30 years	3.330 (1.605~6.910) ***		3.852 (1.654~8.970) **	6.148 (2.551~14.819) ***	3.811 (1.519~9.652) **	5.896 (2.321~14.977) ***	3.040 (0.968~9.555) *	4.059 (1.126~13.553) *			0.487 (0.322~0.736) ***	0.354 (0.198~0.631) ***
**Hazardous exposure**												
Vibration (Yes)		2.306 (0.897~5.927)	0.511 (0.261~0.999)*									
Noise (Yes)	0.387 (0.171~0.875)*		2.481 (0.807~7.626)									
High temperature (Yes)	1.907 (1.052~3.457) *				1.776 (0.898~3.512)					3.291 (0.898~12.056)	0.748 (0.571~0.981) *	
Low temperature (Yes)												
Smoke/fumes/dust (Yes)			0.779 (0.585~1.039)				0.746 (0.548~1.017)					
Chemical substances (Yes)	3.086 (1.061~8.974) *										1.603 (0.962~2.668)	
Fatigue/pain-inducing posture (Yes)					1.772 (0.916~3.428)				5.071 (0.945~27.223) *			
Heavy objects (Yes)	1.961 (1.132~3.396) **		3.322 (1.876~5.885) ***	2.874 (1.499~5.508) ***	2.721 (1.460~5.070) **	1.876 (0.973~3.617)		2.195 (1.091~4.415) *	11.371 (2.224~58.138) **	4.358 (1.189~15.975) *		
Standing posture (Yes)		2.105 (1.090~4.066) *		1.852 (0.909~3.774)								
Sitting posture (Yes)	0.480 (0.253~0.912) *		0.380 (0.202~0.714) **		0.279 (0.124~0.625) **						0.739 (0.551~0.990) *	
Repetitive hand/arm movement (Yes)	3.278 (1.887~5.693) ***	2.289 (1.197~4.378) **	4.254 (2.354~7.696) ***	2.018 (0.989~4.118) *	4.242 (2.183~8.242) ***	2.739 (1.292~5.806) **	2.774 (1.102~6.985) *					
**Safety information** (Not given)	1.758 (1.026~3.015) *	1.892 (1.029~3.479) *	2.799 (1.642~4.770) ***		2.036 (1.097~3.782) *			2.260 (0.973~5.250) *			0.312 (0.229~0.450) ***	0.554 (0.368~0.834) **

* *p* < 0.05, ** *p* < 0.01, *** *p* < 0.001; N: Number of samples; OR: Odds ratio; CI: Confidence interval; Age standardized.

## Data Availability

The data that support the findings of this study are available on request from the Occupational Safety and Health Research Institute, Korea Occupational Safety and Health Agency.

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
