# Peer review of "Differences in the Effects of Work Environment on Health Problems and Satisfaction of Working Condition by Gender: The 6th Korean Working Conditions Survey"

_ijerph, 2023, doi:10.3390/ijerph20196824_

Round 1

Reviewer 1 Report

The authors reported that female farmworkers received less health and safety information and had lower work environment satisfaction compared to male farmworkers from 6th Korean Working Conditions Survey (KWCS) conducted by the Occupational Safety and Health Research Institute (OSHRI) in Korea.

However, some concerns have been raised.

1.     Further, the percentages of male and female workers exposed to “severe noise that requires a loud voice when talking to others” were 11.1% and 10.6%, respectively. These gender gaps were not statistically significant. However, in Table 2, the p value is < 0.000, please revised it and added the true value in Table 2.

2.     In line 204-211, regarding ergonomic risks, there were no significant differences in exposure to “repetitive hand/arm movement” between men (80.5%) and women (81.6%), but exposure to other ergonomic risk factors significantly differed between genders. Women (81.0%) were more exposed to “sitting posture” than men (69.0%), while men (89.2%) were more exposed to “continuous standing posture” than women (84.3%). “Pulling/pushing/moving heavy objects” was more common among men (69.4%) than women (22.0%), while “posture that causes fatigue or pain” was more common among women (70.5%) than men (64.2%). Did not find any Table to describe these results. Please add a Table to describe it.

3.     In line 233-235, gender differences in work environment satisfaction among farmworkers were analyzed (Table 5). The percentage of farmworkers who are satisfied with their work environment was significantly higher in men (77.9%) than women (72.0%). However, in Table 5, can not find the results.

4.     Please revised the results of “3.6. Predictors of work-related health problems and work environment satisfaction among farmworkers by gender.” It is hard to read by readers (Table 6 and Table 7).

5.     The discussion is too longer and please simple to discuss these finding.

Please rewrite the results of 3.6. and the discussion.

Reviewer 2 Report

Review

General Comments:

The manuscript entitled “Differences in the Effects of Work Environment on Health  Problems and Satisfaction of Working Condition by Gender: 3 The 6th Korean Working Conditions Survey” is the result of the research of an interesting topic that implies differences by gender in the farm industry. This research was performed using statistical and data science techniques for assessing the differences between both groups in a wide number of variables regarding specific subjects. In general, the manuscript has high quality research and scientific sounding, although some minor english mistakes. Introduction is very solid and contextual. Materials and methods are well described. The stratified sampling is important for this kind of study where there is a clear imbalance between farm workers' occupations, so it was a good strategy. Also, the way variables were handled (as binary data to deal with string responses) was appropriate and well described. The only “weak point” is that, since this work is about evaluating the difference between both groups (female and male), it would be necessary to explicitly clarify the test used for this evaluation. The P value is used, which is correct, but the kind of test should be mentioned, whether t-test was used or its Mann-Whitney U. For the comparison between different categorical variables, it is clear the use of Chi square from the title, so there is no problem with that.  Results are meticulously described in text, but I believe this kind of research has the need of more suitable visualizations, also it would increase the impact of the paper overall. Discussion is very insightful and the cited references complement the research results. It is interesting the conclusions in which female farmworkers are less satisfied with their work environment than male farmworkers, this could also be influenced by the fact that less female farmworkers are actually paid and also the lack of health and safety-related information. It would be interesting to explore the correlation between this idea. But in general, the conclusions highlight the need for different measures to improve farmworkers' environment according to their gender.   

The manuscript results are worthy of being published, nevertheless I strongly recommend better visualizations, since for this kind of research I consider visualizations as important as the analysis. I suggest visiting the next graph galleries for ideas:

https://seaborn.pydata.org/examples/index.html

https://plotly.com/python/

In conclusion, I recommend the publication of this journal under major revisions regarding better data visualization. 

Specific Comments: 

Line 38 - 41 These sentences look a little bit disconnected with this paragraph, I suggest some paraphrasing to connect this idea with the previous. Or even move these sentences to the paragraph from lines 60 - 70. 

Line 123 I suggest briefly clarifying the criterion for discarting other variables and how the authors selected the relevant features. 

Line 129 I believe (Q) must be modified. 

Line 173 I think would be appropriate to clarify the test performed to evaluate the difference between groups, like the t-test or its Mann-Whitney U version, etc… just as a quick mention. 

Table 6 and 7 design is very confusing. There must be another way to depict predictors OR in a panel of figures or something more appropriate.  

English only requires minor revisions. 

Reviewer 3 Report

The article is of interest in the field of occupational health, however the context is reduced to Korea. The authors must make this context clear, mentioning at the beginning of the introduction that the study is carried out in Korea.

Line 87: especify what gender differences were reported previously.

Methods:

Lines 117-120 show % together with the numbers.

Line 127: add "and health consequences" at the end of the sentence.

Results:

Please describe the total distribution by gender at the beggining of the text.

Discussion: include that the study refers to agricultural workers in Korea (line 335).

Line 355: you are giving an explanation, but this is not certain. Please redo the sentence conditionally.

Line 373: blurred vision is indeed a symptom of organophosphates intoxication

Line 499: what do you mean with "inadequate research"? Please redo the sentense. There is controversy? It is controversial?

Lines 514-516: eventhough the study was done in Korea, results may be extrapolated to other countries with similar reality regarding agricultural environment.

At the end of the discussion, I suggest to add somethig aboput incorporating a gender approach not only in the study of factors related to work in the agricultural sector, but also when developing preventive strategies and occupational health policies.

There are few sentences to improve

Round 2

Reviewer 1 Report

These authors response to my questions.

It is OK in present status.